# Glove-Net: Enhancing Grasp Classification with Multisensory Data and Deep Learning Approach

**DOI:** 10.3390/s24134378

**Published:** 2024-07-05

**Authors:** Subhash Pratap, Jyotindra Narayan, Yoshiyuki Hatta, Kazuaki Ito, Shyamanta M. Hazarika

**Affiliations:** 1Department of Mechanical Engineering, Indian Institute of Technology Guwahati, Guwahati 781039, India; s.m.hazarika@iitg.ac.in; 2Department of Mechanical Engineering, Gifu University, Gifu 501-1193, Japan; hatta.yoshiyuki.b3@f.gifu-u.ac.jp (Y.H.); ito.kazuaki.x5@f.gifu-u.ac.jp (K.I.); 3Department of Computing, Imperial College London, London SW7 2RH, UK; 4Chair of Digital Health, Universität Bayreuth, 95445 Bayreuth, Germany

**Keywords:** data gloves, human grasp, grasp classification, deep learning

## Abstract

Grasp classification is pivotal for understanding human interactions with objects, with wide-ranging applications in robotics, prosthetics, and rehabilitation. This study introduces a novel methodology utilizing a multisensory data glove to capture intricate grasp dynamics, including finger posture bending angles and fingertip forces. Our dataset comprises data collected from 10 participants engaging in grasp trials with 24 objects using the YCB object set. We evaluate classification performance under three scenarios: utilizing grasp posture alone, utilizing grasp force alone, and combining both modalities. We propose *Glove-Net*, a hybrid CNN-BiLSTM architecture for classifying grasp patterns within our dataset, aiming to harness the unique advantages offered by both CNNs and BiLSTM networks. This model seamlessly integrates CNNs’ spatial feature extraction capabilities with the temporal sequence learning strengths inherent in BiLSTM networks, effectively addressing the intricate dependencies present within our grasping data. Our study includes findings from an extensive ablation study aimed at optimizing model configurations and hyperparameters. We quantify and compare the classification accuracy across these scenarios: CNN achieved 88.09%, 69.38%, and 93.51% testing accuracies for posture-only, force-only, and combined data, respectively. LSTM exhibited accuracies of 86.02%, 70.52%, and 92.19% for the same scenarios. Notably, the hybrid CNN-BiLSTM proposed model demonstrated superior performance with accuracies of 90.83%, 73.12%, and 98.75% across the respective scenarios. Through rigorous numerical experimentation, our results underscore the significance of multimodal grasp classification and highlight the efficacy of the proposed hybrid *Glove-Net* architectures in leveraging multisensory data for precise grasp recognition. These insights advance understanding of human–machine interaction and hold promise for diverse real-world applications.

## 1. Introduction

Grasping and manipulation are fundamental functions in robotics, which are characterized by complex interactions between robotic hands and a variety of objects. These interactions are organized into specific grasp types, each necessitating distinct hand configurations and force applications. The accurate understanding and classification of these grasp types are vital for refining robotic manipulation and carry significant implications across diverse fields, including robotics, medical device technology, and biomechanics. These classifications help enhance the dexterity and adaptability of robotic systems, making them more capable of performing nuanced tasks across different environments.

Despite significant advancements in sensor-based grasp classification techniques, robotic systems still face challenges in mastering complex manipulation tasks [1]. A crucial barrier involves the simultaneous interpretation of comprehensive grasp insights during hand–object interactions, encompassing hand postures and forces. These insights are vital for a thorough understanding of manipulation dynamics. Grasp gesture recognition typically embraces two primary methods: vision-based methods (non-wearable) and data gloves (wearable) [1,2,3]. Vision-based methods involve capturing human gestures through cameras and processing image sequences to extract and classify gesture features. Modern vision-based sensors and motion-capture systems, while adept at providing detailed trajectory data, often fail to capture the nuances of physical interactions fully [4,5]. This limitation is especially problematic in tasks requiring precise hand–object interactions, such as unscrewing a bottle cap, where actions like squeezing or twisting might appear visually identical [6,7]. The reliance on visual feedback frequently proves inadequate because of significant occlusions between the hand and the object, severely compromising data reliability [1,8].

On the other hand, data gloves offer a promising solution by directly capturing hand movements, thereby circumventing the occlusion issue [9]. These gloves are equipped with sensors that provide detailed and continuous data on grasp, enabling precise and reliable grasp classification even in scenarios where visual information is obstructed. The potential of data gloves to address occlusion issues and improve data reliability in complex manipulation tasks is significant [1]. A primary difficulty lies in simultaneously processing hand gestures and forces, which is crucial for understanding manipulation dynamics and achieving precise hand–object interactions [10]. This reinforces the argument for the development of multisensory data gloves, which effectively integrate sensor data to overcome the limitations of both methodologies. Integrating force sensors with gesture recognition technologies, particularly through sensor-based data gloves, has shown promising results in enhancing the understanding and classification of hand grasps [10,11]. These wearable technologies offer a comprehensive view of manipulation by capturing both the gestural configuration of the hand and the forces exerted during object interaction. This multimodal data approach improves the precision and functionality of robotic systems and facilitates the development of sophisticated robotic manipulations and advanced prosthetic devices. Data gloves equipped with sensors effectively understand the human grasp, translating these complex movements and pressures into digital data that robotic hands can analyze and replicate. This technology not only enhances human–machine interactions but also aids in developing intelligent prosthetic systems and contributes to the broader field of biomechanics.

Building upon the capabilities of sensor-based data gloves, this research is dedicated to systematically evaluating and comparing the effectiveness of different sensor data types, specifically bending angles and fingertip forces, and various computational models, including CNN, LSTM, and CNN-BiLSTM, in the realm of robotic grasp classification. The primary goal of this study is to identify the most effective combination of sensory data and modeling techniques that enhance the precision and efficiency of grasp classification. We hypothesize that a holistic integration of grasp postures, which provide detailed positional data of the fingers, and fingertip forces, which measure the interaction pressure, will offer a more comprehensive understanding of grasp dynamics. This investigation will assess the performance of each model individually and in combination to determine the optimal configuration for accurate and reliable grasp classification. The findings from this study are expected to advance the fields of robotics and biomechanics significantly, facilitating improvements in robotic grasping and aiding in the design of next-generation rehabilitation devices such as prosthetic hands and exoskeletons that more closely emulate human hand functionality [10,11]. The significant contributions of this research are outlined below:We proposed *Glove-Net*, a hybrid classification model that integrates the strengths of Convolutional Neural Networks (CNN) and Bidirectional Long Short-Term Memory (BiLSTM) networks based on ablation study. This approach leverages CNNs’ ability to extract local features and patterns from gesture data with BiLSTMs’ strength in capturing long-range dependencies and context from past and future states.We investigated the potential of the proposed model for three sets of data, namely, bending angle, fingertip forces, and a combination of both, by evaluating the performance metrics such as accuracy, precision, recall, and F1-score. Later, we compared the proposed model accuracy with the CNN and LSTM models for all three cases.The study confirms that combining data of bending angle and fingertip forces yields better classification accuracy than using these modalities separately. Our test results demonstrate that our model outperforms existing algorithms in accuracy, precision, recall, and F1-score, underscoring the efficacy of integrating multiple sensor data types for robust grasp classification.

## 2. Literature Review

The way we interact with our environment through our hands has been extensively studied and classified. Recent advances in grasp categorization involve detailed analysis of different grasp types and their applications in robotic manipulation. The study of hand grasps has evolved significantly over time. Schlesinger [12] initially identified six fundamental grasp patterns: cylindrical, tip, hook, palmar, spherical, and lateral. Building on this, Napier [13] distinguished between power grasps, which are used for stability and strength, and precision grasps, which are used for dexterity. Cutkosky [14] further refined these categories by considering object shape, size, and non-prehensile postures. Bullock [15] expanded the taxonomy to include subtle within-hand movements, while Lyons [16] introduced geometric classifications like encompass and lateral grasps. In other work on hand grasp categorization, Feix [17] proposed a comprehensive taxonomy of 33 grasp types based on the hand’s configuration and the object’s properties, which is crucial for precision in robotics and prosthetics.

A comprehensive examination of the current literature unveils various methodologies for grasp classification. Traditional methods, grounded in biomechanical analyses, have established the groundwork for comprehending fundamental grasp patterns [18]. Grasp classification has been explored through a range of modalities, encompassing EMG signals [18], visual data [19], and data gloves that capture finger movements [4] and measure fingertip forces [20]. Embedded sensors within data gloves vary in nature and functionality. These sensors generally fall into four categories suitable for hand-related tasks: bend sensors, stretch sensors, inertial measurement units (IMUs), and magnetic sensors [21]. While the majority of current data gloves employed for hand pose modeling rely on bend or stretch sensors, there are instances where gloves incorporate a combination of multiple sensor types. Such data gloves are embedded with tactile and bend sensors to measure the synchronized multimodal grasp information [22,23]. In grasp classification, recent progress integrates wearable sensor data and transmission hardware with intelligent learning algorithms. Developing a classification model stands as a pivotal phase in grasp recognition. This process involves training sample data with appropriate algorithms to create models capable of identifying new data and allowing for continuous improvement through retraining. The choice of model depends on the nature of the data and research goals. Machine learning (ML) algorithms remain prevalent in glove-based gesture recognition. These include methods such as naive Bayesian (NB) [24,25], logistic regression (LR) [26], decision trees (DT) [27], support vector machines (SVMs) [28,29], and K-nearest neighbors (KNNs) [25,27,28]. While the utilization of machine learning with a data glove to classify acquired data into various sign languages is widespread [25,29], only a few studies have applied machine learning techniques for classifying grasps performed using a data glove [27,30] with an average accuracy of 93 percent using SVMs. Researchers [31] conducted a comparative study revealing that SVM exhibited the highest overall accuracy and the lowest sensitivity to training sample sizes, which was followed by RF and k-NN. SVMs stand out in training sample sizes, particularly in tasks like assessing locomotion quality using wearable sensor data. While SVMs require hyperparameter tuning, a grid search algorithm can estimate suitable values. To address challenges like poor generalization, it is crucial to train the model on a representative dataset [32]. Nassour et al. [33] used a sensory glove with linear regression to estimate joint angles and identify 15 gestures with 89.4% accuracy. Chen et al. [27] developed a wearable rehabilitation system that recognized 16 finger gestures using SVM, achieving 93.32% accuracy. Maitre et al. [34] created a data glove prototype that recognized objects in eight daily activities with 95% accuracy using random forest. Lin et al. [35] employed linear regression to classify three hand movements with a high accuracy of 98.0%.

However, in the domain of data gloves, researchers have increasingly turned their attention to applications of deep learning techniques for grasp classification [2,4,36,37,38]. CNNs are recognized as the most prevalent deep learning algorithms, which are characterized by stacked convolutional filters, activation functions, and pooling layers. These components collectively enable the effective extraction of discriminative features from time-series data. CNNs have succeeded in diverse glove-based gesture classification tasks, including sign languages and custom classifications. CNNs were specifically employed to classify hand poses obtained with a data glove using a large-scale tactile dataset, achieving a classification accuracy of 89.4% [39]. A knitted glove was introduced by Lee et al. [38], which is capable of pattern recognition for hand poses. Additionally, they designed a novel CNN model for conducting experiments on hand gesture classification. The experimental results revealed that the proposed CNN structure effectively recognized 10 static hand gestures, achieving classification accuracies ranging from 79% to 97% for different gestures with an average accuracy of 89.5%. Emmanuel et al. [4] pioneered the application of CNN in classifying grasps through piezoelectric data gloves. Experimental data were collected in which each participant performed 30 object grasps based on Schlesinger’s classification method. The findings illustrated that the CNN architecture attained the highest classification accuracy, reaching 88.27%. LSTM, a recurrent neural network (RNN), is tailored to retain long-term dependencies through its memory cell structure, enabling effective grasp classification when trained with suitable models. Tai et al. [40] proposed a new sensor-based continuous hand gesture recognition algorithm using LSTM. Although their experimental results demonstrated the efficacy of the approach, they did not compare it with other contemporary or traditional methods. A data glove designed for real-time dynamic gesture recognition incorporated LSTM neural networks, fully connected layers, and advanced algorithms for the precise localization and recognition of gestures [41]. A data glove equipped with 3D flexible sensors and wristbands utilized a deep feature fusion network to capture detailed gesture information effectively [24]. This method combined multisensor data using a CNN with residual connections and processed long-range dependencies of gestures via LSTM, achieving high precision on the American Sign Language dataset.

Recent studies have demonstrated that hybrid models, which merge the capabilities of CNNs and LSTM networks, significantly enhance the accuracy and robustness of classification tasks such as grasp recognition [42,43,44,45]. CNNs effectively extract spatial features from data but do not capture temporal dependencies, which are vital for understanding sequences of actions or gestures. On the other hand, LSTMs excel in processing time-dependent data but might overlook the spatial complexities that CNNs can detect [46,47]. In this context, Lopez et al. [42] utilized EMG signals for hand gesture recognition, integrating a post-processing algorithm that significantly boosts recognition accuracy by filtering out erroneous predictions. This approach proved more impactful than solely using LSTM, enhancing the CNN model’s accuracy by 41.86% and the CNN-LSTM model’s by 24.77%. Moreover, while the inclusion of LSTM improved recognition by 3.29%, it did so at a high computational cost. The hybrid CNN-LSTM model with post-processing reached an impressive average accuracy of 90.55%, offering a promising direction for future grasp recognition research [42]. Karnam et al. [43] introduced a hybrid CNN-BiLSTM architecture (EMGHandNet) that classifies hand activities using sEMG signals by integrating spatial and bidirectional temporal data. Evaluated across five benchmark datasets, this model achieved superior classification accuracies, notably 91.29% on the BioPatRec DB2 dataset, demonstrating its effectiveness over existing methods and highlighting its potential to advance hand gesture recognition technologies [43]. Wu et al. [44] explored the use of data gloves for dynamic hand gesture recognition in their study, introducing an innovative model, the Attention-based CNN-BiLSTM Network. This model combines CNNs for local feature extraction and BiLSTMs for contextual temporal analysis, which is further enhanced by attention mechanisms to improve recognition accuracy. Across seven dynamic gestures, the hybrid model demonstrated superior performance, achieving a notable accuracy of 95.05% on the test dataset [44].

Modeling force distribution in grasping is essential for applications requiring physical human–robot interaction, such as robotic prosthetic hands. These systems must mimic human-like force patterns for natural interactions [48]. Numerous devices and methods have been developed to study force distribution across different grasps [20,49,50]. Abbasi et al. [48] used a data glove with 17 sensors to identify unique force patterns for each grasp type, creating a robust taxonomy that offers valuable insights for designing robotic hands and grasp controllers. Understanding these patterns is crucial for enhancing robotic and prosthetic systems’ functionality. Integrating force sensory feedback into grasp classification improves accuracy and supports advanced control strategies, leading to more intuitive and effective human–robot interactions [49,50].

Despite the significant advancements in hand gesture recognition through machine learning, deep learning, and hybrid models, previous studies have focused on gesture classification without fully capturing the complete spectrum of grasp characteristics. This underscores a significant research gap: the limited studies of a multimodal grasp classification system integrating both posture and force data. Achieving a genuinely thorough grasp of classification requires the integration of diverse modalities. While gesture recognition provides valuable insights into hand movements, it does not account for the force exerted during these interactions, which is critical for understanding grasp dynamics. Putting force sensors in data gloves could enable the simultaneous measurement of finger bending and the forces applied, offering a more detailed view of how objects are manipulated. This multimodal approach would detect subtle nuances in how different objects are handled, distinguishing between similar gestures that may require different levels of force. Such detailed data are crucial for applications requiring high precision in exoskeletons and prosthetics, where understanding the intensity and stability of a grasp can greatly enhance functionality and user safety. Due to limited research for multimodal grasp data, many potential opportunities regarding wearable sensors and intelligent algorithms remain unexplored. To tackle these challenges, researchers have advocated for a human-inspired strategy known as multimodal learning, which involves learning a task from various sensory modalities [51]. For instance, diverse sensors can capture the same event from different perspectives. Moreover, by incorporating multiple sources of information, multimodal perception enables models to acquire a more resilient understanding of a task than relying solely on unimodal data [36,52]. A thorough understanding of parameters like finger-bending angles and fingertip pressures could enhance grasp recognition and assessment and aid in planning effective rehabilitation strategies.

Based on the above literature gap and related deductions, the authors were inspired to develop a cost-effective 3D-printed flexible multisensory data glove with embedded flex sensors and force sensors positioned strategically on the dorsal and palmar regions of the hand [53]. This glove aimed to gather valuable insights into human grasp regarding posture and force. Our research hypothesis posits that integrating multimodal data through a hybrid model offers a more robust framework for grasp classification. This approach capitalizes on the detailed spatial information captured by CNNs and the temporal insights provided by BiLSTMs, resulting in optimized classification accuracy. By employing this method, we aim to enhance the practical application of such technologies in robotics and prosthetics development, ultimately leading to more nuanced and effective human–machine interactions.

The structure of the rest of this paper is as follows. Section 3 discusses the methodology, including the 3D-printed multisensory data glove, data collection, and our hybrid deep learning approach featuring CNNs and BiLSTM within the *Glove-Net* model. Section 4 presents the classification results using angle, force, and combined data and analyzes model performance. Section 5 discusses the findings and their implications for robotic grasp classification. Section 6 concludes the paper, summarizing key contributions and suggesting future research directions.

## 3. Methodology

### 3.1. The 3D-Printed Multisensory Data Glove

The data glove is designed to enhance grasp classification by integrating precise sensors capable of capturing hand movement patterns and fingertip forces. Fabricated from Raise3D Premium TPU-95A filament, the glove combines elasticity, resilience, and durability, ensuring user comfort and adaptability across various applications, including virtual reality and clinical diagnostics. It features strategically placed sensors, including Finger Tactile Pressure Sensors (FingerTPSs) and flex sensors, which are crucial for the accurate data acquisition of finger movements and applied forces. These sensors are embedded in a lightweight structure that conforms to different hand shapes, facilitating natural movement and ease of use. The complete system, supported by an Arduino-based instrumentation board, captures and displays real-time data effectively, serving both professional and recreational purposes comprehensively. The design of the data glove in this work is centered around three pivotal considerations: Sensor Integration, Wearability and User Comfort, and Generic Design for multiple users. Detailed discussions on the glove’s design considerations and functionalities are elaborated in the authors’ prior research [53], providing a deeper insight into its technical and practical implications. The complete setup of the 3D-printed fabricated data glove is illustrated in Figure 1.

The grasp posture measurement facilitated by the data glove involves flex sensors embedded within, which provide a generalized measurement of joint flexion by capturing the overall flexion of the MCP, DIP, and PIP joints combined. These sensors account for the collective movement patterns rather than the specific angles of individual finger flexions, ensuring comprehensive data acquisition. The sensors’ design allows for real-time monitoring and the accurate capturing of dynamic hand movements, which are critical for evaluating hand functionality during various tasks. The Arduino-based electronic hardware configuration further enhances this system’s robustness, which processes the sensor data to provide detailed and reliable output. For detailed insights into the grasp posture measurement and sensor configurations, please refer to the authors’ previous work [53]. Additionally, fingertip force measurement employs capacitive Finger TPS sensors, which detect subtle pressure changes and grasp forces with high sensitivity and repeatability. These sensors, crucial for assessing the tactile feedback during object manipulation, provide valuable data for understanding interaction dynamics. Their performance and technical specifications are detailed further in the authors’ previous work, where their application in enhancing grasp recognition accuracy is comprehensively explored [53].

### 3.2. Data Collection

The sensorized glove developed is designed to detect finger flexion/extension and fingertip forces while grasping household objects from the Yale–CMU–Berkeley (YCB) set [54]. An experimental setup is tailored for a specific grasping task, which is benchmarked by the Anthropomorphic Hand Assessment Protocol (AHAP) utilizing 24 objects from the publicly accessible YCB object and model set, ensuring potential replication in future studies [55]. Grasping capability involves securely grasping various daily-life objects and sustaining a stable grip, including eight different grasp types (GTs): pulp pinch (PP), lateral pinch (LP), diagonal volar grip (DVG), cylindrical grip (CG), extension grip (EG), tripod pinch (TP), spherical grip (SG), and hook grip (H) [17]. The set includes 24 tasks encompassing these GTs, which were chosen based on research in human grasp analysis, prosthetics, and rehabilitation [17]. A selection of three objects from the YCB collection represents each grasp type, accounting for variations in size, shape, weight, texture, and rigidity, totaling 24 objects covering food, kitchen, tool, shape, and task categories [55]. Figure 2 illustrates the eight distinct grasp types incorporated into this study. The specifications of each object are explained in the authors’ previous work [53].

Ten healthy subjects, aged 25-45, including six males and four females, were recruited for the study. They self-reported good hand health, without any pain, injury, or disease such as arthritis. The experimental procedures followed ethical guidelines outlined in [56], prioritizing subject well-being and data credibility. Approval was obtained from the Institute Human Ethics Committee (Reference: IHEC/SH/1/2020). All participants were right-handed and showed no complications related to cognition or upper extremity function, aligning with criteria for individuals with hemiparesis.

Establishing clear experimental protocols and benchmarks is essential due to the diverse interests and evolving nature of manipulation research, making it challenging to craft adaptable task descriptions. The experiment considered factors such as shape, size, weight, grasp gesture, and stiffness to accommodate various manipulation types. Subjects conducted repeated grasp trials with various grasp types using different everyday objects, with the camera positioned 60 cm from the object, capturing 30-second grasping trials. Subjects began with their gloved hand resting on the table and then grasped and elevated the object to a 15 cm height, maintaining a stable hold for 20 s. Data collection divided each trial into four phases: Approaching, Grasping, Lifting, and Holding. The experimental setup is detailed in Figure 3.

During trials, some fingers made contact with the object without yielding discernible sensor readings, which was likely due to grasping by finger pads and the palm without involving fingertips where force sensors are located. To avoid this, participants executed natural grasps without additional constraints, resulting in diverse grasp postures recorded for each object. In .csv format, the dataset contains grasp force (in newtons) and finger-bending angles (in degrees) for the Thumb, Index, Middle, Ring, and Pinky fingers in each trial. Tactile and bending angle data consist of 5 channels and 1200 steps within 30 s.

The subjects performed 10 trials per object, resulting in a total of 2400 samples (10 subjects × 8 grasp types × 3 objects per grasp type × 10 trials). Each trial was recorded over 30 s at a frequency of 40 Hz, yielding 1200 data points per trial, which gives a total of 2,880,000 data points (2400 trials × 1200 data points per trial). This extensive, high-resolution dataset supports the effective training of our CNN-BiLSTM model, ensuring an accurate classification of grasp types and robust generalization. The high testing accuracy confirms the dataset’s adequacy for reliable neural network training.

### 3.3. The Deep Learnig Classification Approach

#### 3.3.1. Convolutional Neural Networks (CNNs)

A Convolutional Neural Network (CNN) is a deep learning model designed for processing data with grid-like structures, such as images or sequences. CNNs excel at feature extraction and pattern recognition by applying convolutional layers to capture spatial hierarchies in the data. In a 1D-CNN, convolution layers use filters (kernels) on input data to create feature maps, performing element-wise multiplications and summations to detect local patterns and relationships like those between bending angles and fingertip forces. Pooling layers, such as MaxPooling1D, down-sample the feature maps by selecting the maximum value within a window, reducing dimensionality and computational complexity while retaining significant features. This makes feature detection invariant to small translations in the input data. dropout layers prevent overfitting by randomly setting input units to zero during training, ensuring the network learns robust, generalized features rather than memorizing the training data. Activation functions like the Rectified Linear Unit (ReLU) introduce non-linearity, helping the network learn complex patterns by outputting zero for negative inputs and the input value for positive inputs, speeding up convergence. Key hyperparameters in a 1D-CNN include the number of CNN layers, the number of neurons (filters) in each layer, the filter size (kernel size), and the subsampling factor (pooling size). The network’s depth, determined by the number of convolution layers and neurons per layer, influences its ability to capture various features, while filter size and pooling size control the scope and degree of down-sampling.

Despite the effectiveness of 1D-CNNs in spatial feature extraction, they have limitations in capturing temporal dependencies and long-range interactions within sequential data. This limitation is crucial for our grasp classification task, as the temporal dynamics and sequential nature of the bending angles and fingertip forces are essential for accurately identifying grasp types. To address this, one can combine the CNN with a Bidirectional Long Short-Term Memory (BiLSTM) network.

#### 3.3.2. Bidirectional Long Short-Term Memory (BiLSTM)

In contrast to 1D-CNNs, which excel at spatial feature extraction, the BiLSTM network effectively learns temporal dependencies and long-range interactions within sequential data. BiLSTM is a type of recurrent neural network (RNN) that processes sequential data in both forward and backward directions, combining the power of LSTM with bidirectional processing. This enables the model to capture both past and future context of the input sequence, making it highly effective for grasp classification using our dataset. To understand BiLSTM in the context of grasp classification, let us break down its components and functionality:LSTM addresses traditional RNNs’ limitations in capturing long-term dependencies in sequential data by introducing memory cells and gating mechanisms. These allow LSTMs to selectively retain and forget information over time, storing information for extended durations. This is essential for grasp classification, where the relationship between bending angles and fingertip forces over time is critical.Bidirectional processing enhances RNNs by processing input sequences simultaneously in both directions using two LSTM layers: one for the forward direction and one for the backward direction. Each layer maintains its hidden states and memory cells (see Figure 4), ensuring the model captures the full context of the grasping sequence.During the forward pass, the input sequence (bending angles and fingertip forces) is fed into the forward LSTM layer from the first to the last time step. The forward LSTM computes its hidden state and updates its memory cell at each step based on the current input and previous states. Simultaneously, the backward LSTM processes the sequence in reverse, from the last to the first time step, capturing future information during the backward pass.Once the forward and backward passes are complete, the hidden states from both LSTM layers are combined at each time step, either by concatenation or another transformation. This combined information provides a richer understanding of the sequence, capturing dependencies from both past and future time steps.The benefit of BiLSTM for grasp classification is capturing context before and after a specific time step. By considering both past and future information, BiLSTM captures richer dependencies in the input sequence of bending angles and fingertip forces, leading to more accurate grasp classification.

The grasp classification architecture includes several key components. The input sequence consists of vectors representing data points like bending angles and fingertip forces for each finger during a grasping instance. The core component is the BiLSTM layer, which has two LSTM layers: one processes the input sequence forward, and the other processes it backward, each with its parameters. The output of the BiLSTM layer combines the hidden states from both directions at each time step. This output is passed through a fully connected layer and a softmax activation for grasp classification to obtain class probabilities for each grasp type.

Combining CNN and BiLSTM, the architecture captures spatial and temporal features in the grasping data, enhancing classification performance. To improve its capabilities, the BiLSTM can be extended with additional layers, such as fully connected layers. Integrating BiLSTM with CNN addresses the limitations of CNNs in capturing temporal dependencies, providing a robust solution for grasp classification.

#### 3.3.3. The Proposed *Glove-Net* Model

To leverage the strengths of both Convolutional Neural Networks (CNNs) and Bidirectional Long Short-Term Memory (BiLSTM) networks, we propose the *Glove-Net* model as a hybrid CNN-BiLSTM architecture for grasp classification of our dataset. This model effectively combines the spatial feature extraction capabilities of CNNs with the temporal sequence learning strengths of BiLSTM networks, addressing the complex dependencies within our grasping data. With an iterative selection of learning rates and 50 training epochs, the model balances convergence speed and performance optimization, ensuring robust grasp classification. Furthermore, the dataset is strategically divided into three subsets: training, testing, and validation. The training set is used to train the model, the testing set is used to evaluate the model’s performance on unseen data, and the validation set is used to tune hyperparameters and prevent overfitting. During the initial training and evaluation of the Glove-Net model, the dataset was divided into training (80%), validation (10%), and test (10%) sets. This three-set division allowed for effective model training and hyperparameter tuning using the training and validation sets, while the test set was reserved for final performance evaluation.

Fusing CNN and BiLSTM enables the model to capture both spatial and temporal features in the grasping data. The fusion process occurs in three main stages: (1) *Feature Extraction with CNN:* The input grasp data are processed through convolutional layers, which apply filters to detect spatial patterns, resulting in a set of feature maps. (2) *Temporal Processing with BiLSTM:* These feature maps are passed to the BiLSTM layers, which process them bidirectionally to capture temporal relationships, producing a sequence of enriched feature vectors. (3) *Integration and Classification:* The feature vectors are flattened and fed into dense layers that learn high-level representations of the grasp actions, culminating in a softmax layer that classifies the grasp type.

The proposed *Glove-Net* model architecture for grasp classification integrates the strengths of both Convolutional Neural Networks (CNNs) and Bidirectional Long Short-Term Memory (BiLSTM) networks. The model begins with a 1D convolutional layer, comprising 64 filters with a kernel size of 2, which effectively captures local spatial features within the bending angles and fingertip forces of the grasping data. This is followed by a max-pooling layer with a pool size of 2, which reduces the dimensionality of the feature maps, retaining essential features while reducing computational complexity. The model’s core is a bidirectional LSTM layer with 64 units, which processes the sequence data in both forward and backward directions, capturing comprehensive temporal dependencies from past and future time steps. The output from the BiLSTM layer is then flattened into a 1D feature vector, which is further transformed by a dense layer with 32 units and ReLU activation, learning high-level representations of the grasping data. A dropout layer with a 50% dropout rate is included to prevent overfitting, promoting robust feature learning. The final output layer, a dense layer with 8 units and a softmax activation function, classifies the grasping instances into one of the eight grasp types.

The model architecture employed in this research underwent a meticulous refinement process through an extensive ablation study to identify optimal configurations and hyperparameters for precise and comprehensive grasp classification on our dataset. We performed a grid search to explore a predefined set of hyperparameters. This involved evaluating all possible combinations of selected values for each parameter. This study systematically explored various parameter values, encompassing the number of Conv1D filters, Conv1D kernel sizes, LSTM units, and dense layer sizes, as displayed in Table 1. Specifically, we considered Conv1D filters of 32 and 64, Conv1D kernel sizes of 2 and 3, LSTM units of 32 and 64, dense units of 32 and 64, softmax units of 8, and dropout rates of 0.1 and 0.5. By systematically varying these parameters and assessing their impact on classification performance, we tailored the model architecture to effectively capture the intricate nuances of grasp patterns within our dataset, ultimately achieving superior classification accuracy.

The primary objective of our ablation study was to pinpoint the CNN-BiLSTM configuration yielding the highest accuracy in grasp classification. After evaluating 32 configurations, classifier C22, illustrated in Figure 5, emerged with the highest average classification accuracy. Consequently, C22 was designated as the optimal CNN-BiLSTM configuration for our study. This rigorous analysis was pivotal in ensuring that our CNN-BiLSTM algorithm was equipped with the most effective parameters, enhancing its accuracy in classifying grasp patterns.

### 3.4. Performance Matrices

To evaluate the performance of our proposed hybrid CNN-BiLSTM model for grasp classification, we conducted experiments using robust computational resources and software platforms. The software platforms and libraries used in our experiments were Python 3.8, Keras with TensorFlow backend (version 2.4.0) for deep learning, Pandas (version 1.2.3) and NumPy (version 1.19.2) for data manipulation and analysis, Matplotlib (version 3.3.4) and Seaborn (version 0.11.1) for visualization, and Scikit-learn (version 0.24.1) for evaluation metrics.

To comprehensively evaluate the performance of the grasp classification model, we employed the following evaluation metrics: accuracy, F1-score, and confusion matrix. Accuracy measures the proportion of correctly classified instances, providing a basic measure of the model’s overall performance. The F1-score, the harmonic mean of precision and recall, balances these two metrics and is especially useful for imbalanced datasets. The confusion matrix provides a detailed breakdown of correct and incorrect classifications for each grasp type, helping to understand the errors the model makes and identify specific classes that are harder to classify correctly. By employing these evaluation metrics, we can comprehensively assess the performance of our proposed CNN-BiLSTM model for grasp classification, allowing us to understand the strengths and weaknesses of the model and ensuring a thorough evaluation of its effectiveness in classifying the different grasp types in our dataset.

## 4. Results

This section presents a comprehensive analysis of the proposed CNN-BiLSTM (*Glove-Net*) model’s classification performance using three different data type scenarios: angle data, force data, and combined angle and force data. By evaluating key performance metrics such as accuracy, precision, recall, and F1-score, we aim to compare and contrast the efficacy of each data type in grasp classification. Confusion matrices are included to provide a detailed breakdown of model predictions for each grasp type. Additionally, we analyze the validation accuracy curves to assess the generalization capabilities of the models. Comparative graphs are provided to visualize the differences in performance across the various models. This section highlights the advantages of using multimodal combined data and hybrid deep learning model for achieving higher accuracy and reliability in robotic grasp classification.

### 4.1. Classification with Angle Data

The CNN-BiLSTM model was trained using only the angle data from the dataset, which focuses on the bending angles of each finger. The model achieved an overall accuracy of 90.83%. This indicates a strong performance in classifying different grasp types based on finger-bending patterns alone. As detailed in Table 2, the model demonstrated high precision, recall, and F1-scores across most grasp types. For instance, it achieved a precision of 1.00 for both DVG and SG, indicating that the model made very few false positive predictions for these grasp types. Furthermore, the recall for LP, EG, and SG was also perfect at 1.00, suggesting that the model successfully identified all these grasps within the test set.

The confusion matrix for the angle data, depicted in Figure 6, provides a detailed breakdown of the model’s prediction performance for each grasp type. Each cell in the matrix represents the number of times a predicted grasp type matched the actual grasp type with diagonal elements indicating correct predictions and off-diagonal elements representing misclassifications. The cells are color-coded to reflect the density of the values: darker green indicates higher correct predictions, while darker red indicates higher misclassification. The values in the confusion matrix are normalized to reflect the percentage accuracy for each class. The matrix shows that the model correctly classified PP with an average of 5.07 correct predictions, although it misclassified some instances as DVG with 2.60. This translates to an accuracy of 63.33% for PP. LP achieved a perfect classification score of 8.00, reflecting the model’s high accuracy for this grasp type at 100%. Similarly, DVG and CG showed strong performance, with most predictions falling on the diagonal at 7.27 and 7.00, respectively, corresponding to accuracies of 90.83% and 87.50%.

However, some misclassifications were noted. For example, a few instances of PP were misclassified as DVG, with a normalized value of 2.60, contributing to a misclassification rate of approximately 32.50%. CG had some misclassifications into PP with a normalized value of 1.00, reflecting a misclassification rate of 12.50%. Despite these misclassifications, the model’s overall performance using angle data was robust, demonstrating high precision, recall, and F1-scores for most grasp types. These results highlight the model’s ability to effectively leverage finger-bending angles for accurate grasp classification, although some types were more challenging to distinguish than others. Nonetheless, the performance could potentially be enhanced by integrating additional data types.

### 4.2. Classification with Force Data

The CNN-BiLSTM model was further trained using only the force data, which includes the fingertip forces exerted during grasping. The model achieved an overall accuracy of 73.12%. While this performance is reasonable, it indicates that force data alone may not be sufficient for achieving high classification accuracy across all grasp types. As detailed in Table 3, the model exhibited varying precision, recall, and F1-scores for different grasp types. For instance, it achieved a high precision of 0.96 for lateral pinch (LP), indicating a low rate of false positive predictions for this grasp type. However, the recall for pulp pinch (PP) was relatively low at 0.56, suggesting that many instances of this grasp were not accurately identified.

The confusion matrix for the force data, shown in Figure 7, provides insights into the model’s performance in predicting each grasp type based solely on force data.

From the matrix, it is clear that the model correctly classified PP with an average of 4.50 correct predictions, but it misclassified instances into other grasp types, such as 1.82 into DVG. This translates to an accuracy of 55.83% for PP. LP had a correct classification count of 5.38 but showed misclassifications into SG with 1.21, corresponding to an accuracy of 66.67%. DVG achieved 6.45 correct classifications, indicating a strong performance with an accuracy of 80%, although some misclassifications into PP with 0.81 (light red) were noted. However, the model faced significant challenges with certain grasp types. EG had an accuracy of 63.33% with several misclassifications into other types, such as 2.35 into CG. SG achieved an accuracy of 65%, indicating some difficulties in accurately identifying this grasp type, with misclassifications into DVG and H. Despite these challenges, the model exhibited strong performance in certain areas. For instance, TP achieved a near-perfect classification accuracy of 99.17%, demonstrating the model’s ability to leverage force data effectively for this particular grasp type. However, the overall performance, as indicated by the precision, recall, and F1-scores, was generally lower compared to the model trained on angle data. These results suggest that while force data provide valuable information for grasp classification, it may not be sufficient on its own for achieving high accuracy across all grasp types.

Overall, the performance of the CNN-BiLSTM model trained on force data alone underscores the need for additional data types to improve classification accuracy. This observation leads to the hypothesis that integrating angle and force data could potentially enhance the classification performance by leveraging the strengths of both data types.

### 4.3. Classification with Combined Angle and Force Data

Finally, the proposed *Glove-Net* model was trained using the combined angle and force data, leveraging the complementary information from both data types to improve classification accuracy. This multimodal approach aimed to provide a more comprehensive representation of the grasp types, integrating the strengths of both angle and force data. The model achieved an overall accuracy of 98.75%, indicating a significant improvement over the models trained with angle or force data alone. As detailed in Table 4, the combined model demonstrated exceptional performance across all grasp types with precision, recall, and F1-scores consistently high. For instance, the model achieved perfect scores (1.00) in precision and recall for DVG, TP, and SG, reflecting its robust capability to correctly identify these grasp types with no false positives or false negatives.

The confusion matrix for the combined data, illustrated in Figure 8, showcases the model’s prediction performance for each grasp type. From the matrix, it is clear that the model correctly classified PP with an average of 7.93 correct predictions, misclassifying only a minimal number of instances, translating to an accuracy of 99.17%. LP achieved a perfect classification score of 8.00, reflecting a 100% accuracy. DVG and CG showed strong performance, with most predictions falling on the diagonal at 7.73 and 7.73, respectively, corresponding to accuracies of 96.67%. However, the model faced minimal challenges with certain grasp types. For instance, CG had a minor misclassification into PP with a normalized value of 0.27, but this did not significantly impact its overall accuracy of 96.67%. EG, TP, and SG achieved high accuracies of 99.17% and 100%, demonstrating the model’s effective use of combined data for accurate grasp classification. The overall high performance across all metrics and grasp types underscores the efficacy of combining multimodal data for grasp classification.

The comparative performance of the models trained on angle data, force data, and combined data is further illustrated in Figure 9. This figure shows the precision, recall, and F1-score for each grasp type across the three models, highlighting the superior performance of the combined data model. The validation accuracy comparison across epochs for the three models, shown in Figure 10, further supports this conclusion with the combined data model consistently achieving higher accuracy than the individual angle and force data models. In summary, the proposed CNN-BiLSTM *Glove-Net* model trained on the combined angle and force data significantly outperformed the models trained on individual data types. This demonstrates the advantage of integrating multiple modalities to leverage complementary information, ultimately enhancing the robustness and accuracy of grasp classification. The findings strongly suggest that a multimodal approach is highly effective for this application, providing a more accurate and reliable classification framework.

### 4.4. Train–Test Split Comparative Analysis

To further validate these findings, we performed additional experiments using different train–test split ratios across three data types: angle data, force data, and combined data. Throughout these additional experiments, the validation split ratio is kept at 10%. Each experiment was repeated ten times to ensure robustness, and the results were averaged to compute the mean accuracy and standard deviation. This analysis aimed to determine how the proportion of training data affects the model’s accuracy and to compare the performance across the different data modalities. The results of this evaluation are summarized in Table 5. For each data type, the model was trained and evaluated ten times for each train–test split ratio to ensure the robustness of the results. The mean accuracy and standard deviation of accuracy were calculated across these iterations.

The results indicate that the model’s performance improves with an increased proportion of training data, as evidenced by the higher mean accuracies observed for the 80:10 train–test split ratio across all data types. Specifically, the model trained on combined data achieved the highest mean accuracy of 98.75% with a standard deviation of 0.80% for the 80:10 split, highlighting both high accuracy and low variability. This underscores the superior performance and reliability of the multimodal approach, which leverages both angle and force data for enhanced classification accuracy. In comparison, the model trained solely on angle data achieved a mean accuracy of 90.83% with a standard deviation of 0.37% for the 80:10 split. While this performance is commendable, it is significantly lower than that of the combined data model. The model trained on force data alone exhibited the lowest mean accuracy, with a maximum of 73.12% and a standard deviation of 1.88% for the 80:10 split, underscoring the limitations of using force data in isolation for grasp classification. These findings support the hypothesis that integrating multiple data modalities can significantly enhance the performance of grasp classification models. The combined data model consistently outperformed the individual data models across all train–test split ratios, demonstrating the efficacy of a multimodal approach in capturing the complexities of human grasping patterns. Figure 11 provides a comparison bar plot of mean accuracies with error bars representing standard deviations. This plot clearly shows the superior performance of the combined data model across all train–test split ratios, featuring higher accuracies and lower variability compared to models trained on angle or force data alone. Notably, as the training proportion increases from 60% to 80%, the improvement in accuracy is most pronounced for the combined data model, further emphasizing its robustness and efficacy.

### 4.5. Comparative Analysis of Model Performance

In this section, we compared the performance of the proposed CNN-BiLSTM model with contrast models (CNN and LSTM) utilizing angle only, force only, and combined angle and force datasets (Figure 12). The CNN-BiLSTM model trained on combined data achieved the highest performance with a training accuracy of 99.04% and a testing accuracy of 98.75%. This superior performance underscores the efficacy of integrating multimodal data and leveraging the strengths of both CNN and BiLSTM architectures. In contrast, models trained on individual data types, such as CNN-Force and LSTM-Force, exhibited lower accuracies, with the CNN-Force model achieving 73.14% training and 69.38% testing accuracy, while the LSTM-Force model achieved 72.34% training and 70.52% testing accuracy. The models trained on angle data alone performed better than those trained on force data yet still fell short of the combined data models. Specifically, the CNN-Angle model achieved 90.93% training and 88.09% testing accuracy, while the LSTM-Angle model achieved 91.50% training and 86.03% testing accuracy. The CNN-Combined and LSTM-Combined models also demonstrated significant improvements with accuracies of 93.95%/93.51% and 95.92%/92.19% for training/testing, respectively. These results collectively validate the superiority of the combined multimodal data approach for enhancing classification accuracy and reliability in grasp classification tasks.

Figure 12 clearly illustrates the superior performance of the combined data model across all train–test split ratios, with notably higher accuracies and lower variability than the models trained on angle data and force data alone. This comprehensive evaluation underscores the importance of combining multimodal data for grasp classification, providing a compelling case for integrating angle and force data to achieve higher classification accuracy and reliability. The results validate the superior performance of the CNN-BiLSTM model with combined data and highlight the potential for further advancements in robotic grasping applications. The consistent improvement in performance with increasing training data proportions further reinforces the robustness of the multisensory approach.

## 5. Discussion

The results of our study provide significant insights into the efficacy of using a hybrid CNN-BiLSTM model for grasp classification, particularly when leveraging a multimodal dataset comprising both angle and force data. The comparative analysis of classification performance across different data types and models underscores the importance of multimodal data integration. The CNN-BiLSTM model, when trained on combined angle and force data, demonstrated the highest classification accuracy with a mean testing accuracy of 98.75%. This superior performance can be attributed to the model’s ability to capture both spatial and temporal features inherent in the multimodal dataset. While effective, the standalone CNN and LSTM models showed limitations when trained on a single data type. The CNN model trained on angle data achieved a testing accuracy of 88.09%, indicating that angle data alone provide substantial information for grasp classification. However, the performance of the CNN model trained on force data was significantly lower, with a testing accuracy of 69.38%, highlighting the insufficiency of using force data alone for accurate classification. The LSTM models, designed to capture temporal dependencies, performed better than their CNN counterparts when trained on angle data with a testing accuracy of 86.03%. The LSTM model trained on force data, however, still lagged, achieving a testing accuracy of 70.52%. These results indicate that while temporal features are important, they are not sufficient without the complementary spatial features provided by angle data.

The hybrid CNN-BiLSTM model’s ability to combine the strengths of both CNNs and LSTMs is evident from its performance. The CNN component effectively extracts spatial features from the raw sensor data, while the BiLSTM component captures the temporal dependencies inherent in sequential grasp patterns. This combination allows the hybrid model to achieve higher classification accuracy and reliability, as demonstrated by its performance on the combined dataset. The results highlight the model’s robustness and generalization capabilities, particularly in handling the complexities of human grasp patterns. The hybrid model’s superior performance across different train–test split ratios further validates its efficiency and effectiveness. Integrating multimodal data significantly enhances the model’s ability to classify grasps accurately, leveraging the complementary information provided by both angle and force data.

To evaluate the performance of our proposed CNN-BiLSTM (*Glove-Net*) model against other classification algorithms, we conducted a comparative analysis using the same dataset. Table 6 presents the comparison of various models, including traditional machine learning algorithms and deep learning networks. The results demonstrate that the Glove-Net model outperforms all other methods, achieving the highest accuracy of 98.75%. This highlights the effectiveness of our hybrid approach in integrating spatial and temporal features for precise grasp recognition.

Furthermore, we compared the classification accuracy of our proposed CNN-BiLSTM (*Glove-Net*) network with various state-of-the-art algorithms as summarized in Table 7. Our analysis shows that the *Glove-Net* model outperforms existing methods in recognizing different grasp types, achieving a higher classification accuracy. This superior performance demonstrates the effectiveness and robustness of our hybrid approach in leveraging both spatial and temporal features of grasp data, setting a new benchmark in grasp recognition tasks.

The findings of this study have broader significance and implications for the fields of orthotics and prosthetics. In orthotics, accurately classifying grasps using a hybrid model trained on multimodal data can lead to more advanced and adaptable rehabilitation systems. The enhanced grasp classification capabilities in prosthetics can contribute to developing more intuitive and responsive devices for amputee users. By incorporating multimodal sensory feedback, these devices can provide users with a more natural and functional experience, closely mimicking the capabilities of the human hand [57]. Furthermore, by accurately monitoring and classifying hand movements, therapists can design more effective rehabilitation protocols tailored to individual needs, ultimately improving patient outcomes. However, the dataset used in this study, while comprehensive, may not capture the full range of real-world grasp variations, limiting the model’s ability to generalize to novel or untrained grasp types, especially in dynamic environments. Additionally, the model’s performance needs further validation in uncontrolled settings with diverse object properties and subjects with varied neurological conditions. Future research should expand the dataset to include a wider array of grasp types and conditions and explore the integration of continuous human manipulation tasks and additional sensor modalities, such as EEG and EMG data, to enhance the model’s generalization and applicability.

## 6. Conclusions and Future Work

This study presents a significant contribution to grasp classification by demonstrating the superior performance of a hybrid CNN-BiLSTM model, *Glove-Net*, using a multisensory data glove. The glove captures finger-bending angles and fingertip forces, providing a comprehensive dataset that enhances classification accuracy. The hybrid model achieved a mean testing accuracy of 98.75%, significantly outperforming standalone CNN and LSTM models. This underscores the effectiveness of integrating multiple data modalities for improved grasp classification. This study highlights the potential of hybrid neural network architectures in leveraging multimodal data for robust grasp classification. The integration of angle and force data offers a comprehensive understanding of grasp dynamics, paving the way for significant advancements in robotic manipulation, prosthetics, and rehabilitation. In future studies, along with other sensory data, the real-time control of orthosis based on different manipulation tasks will be carried out.

## Figures and Tables

**Figure 1 sensors-24-04378-f001:**
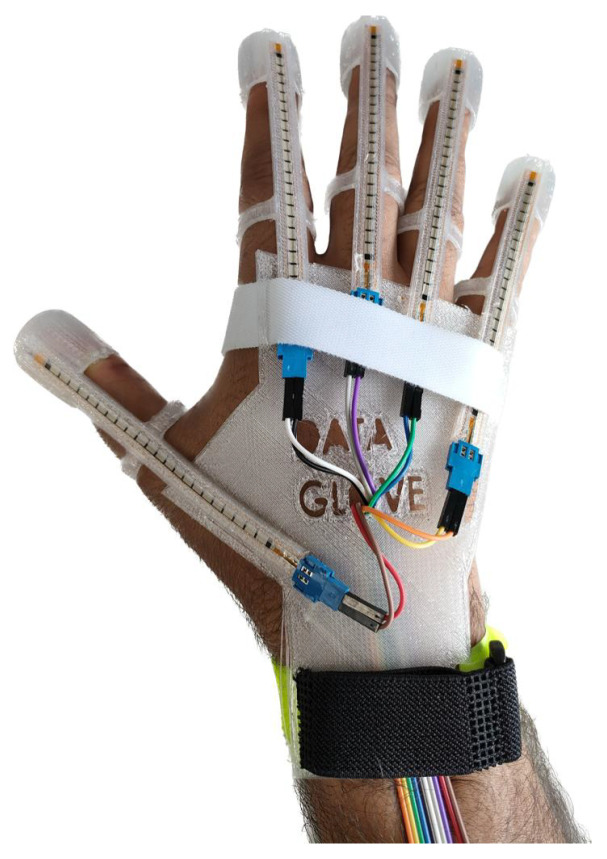
The 3D-printed data glove setup fabricated with the sensors.

**Figure 2 sensors-24-04378-f002:**
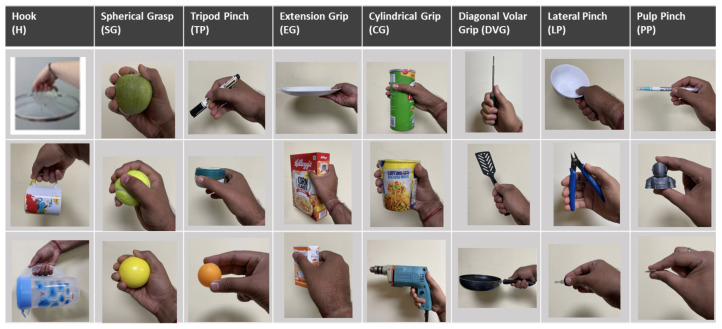
Eight grasp types with three objects from YCB set of objects used in DLAs [53].

**Figure 3 sensors-24-04378-f003:**
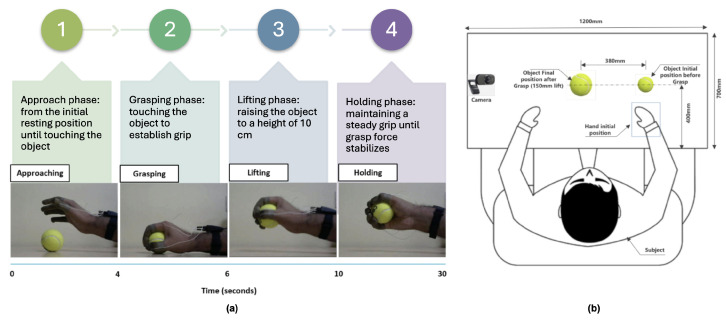
(**a**) Experimental time diagram. (**b**) Schematic diagram of the experimental setup.

**Figure 4 sensors-24-04378-f004:**
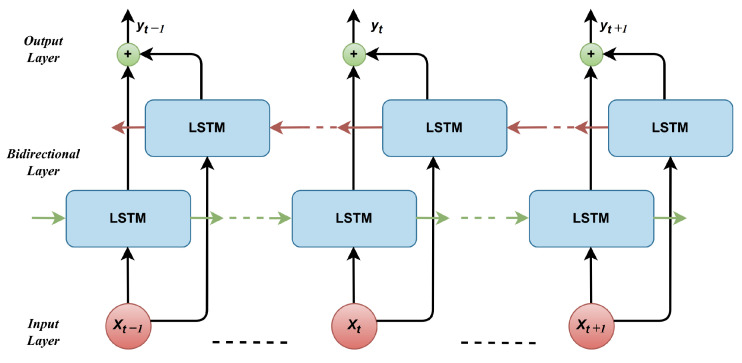
Schematic diagram of BiLSTM architecture.

**Figure 5 sensors-24-04378-f005:**
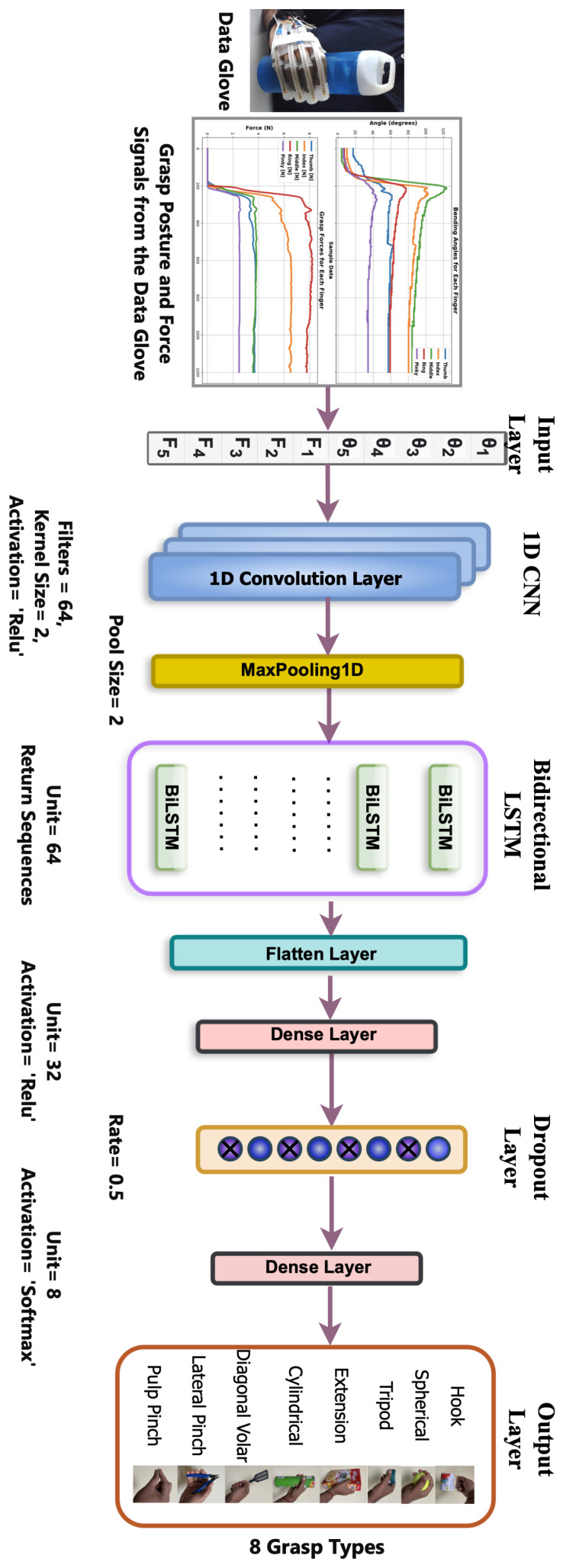
The proposed 1D CNN-BiLSTM Glove-Net classification model architecture.

**Figure 6 sensors-24-04378-f006:**
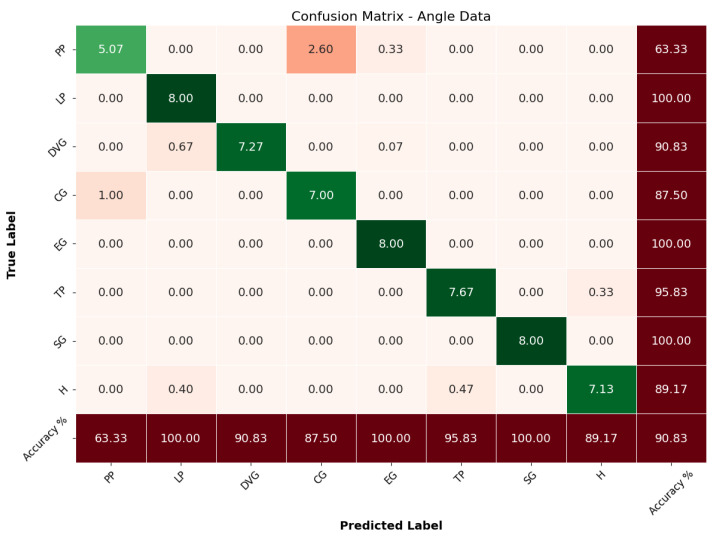
Confusion matrix angle.

**Figure 7 sensors-24-04378-f007:**
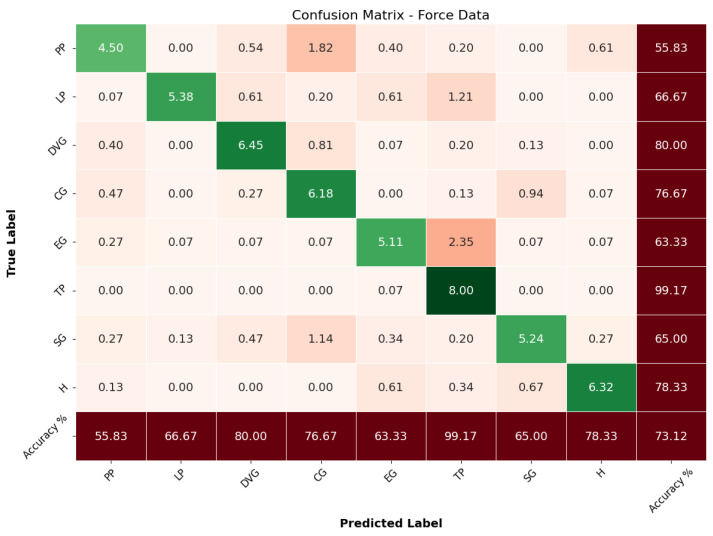
Confusion matrix force.

**Figure 8 sensors-24-04378-f008:**
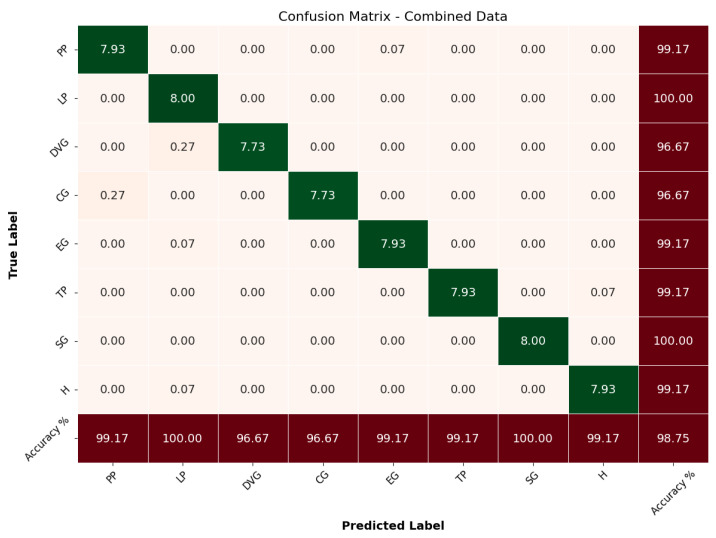
Confusion matrix combined data.

**Figure 9 sensors-24-04378-f009:**
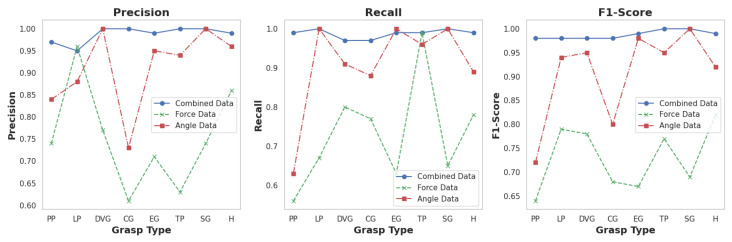
Evaluation matrices (precision, recall and F1-score) comparision.

**Figure 10 sensors-24-04378-f010:**
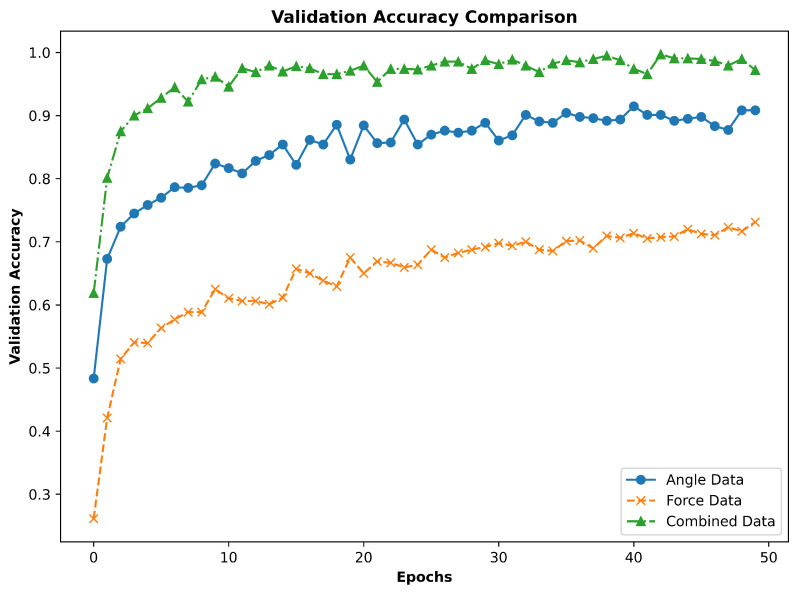
Validation accuracy curves comparision for the three classification scenarios.

**Figure 11 sensors-24-04378-f011:**
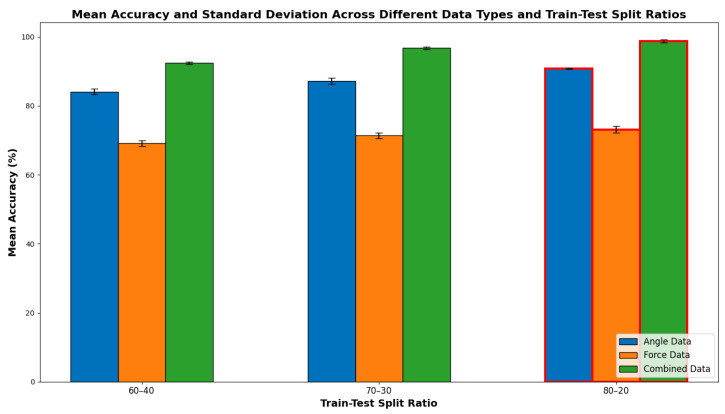
Training and testing accuracy comparison across different models.

**Figure 12 sensors-24-04378-f012:**
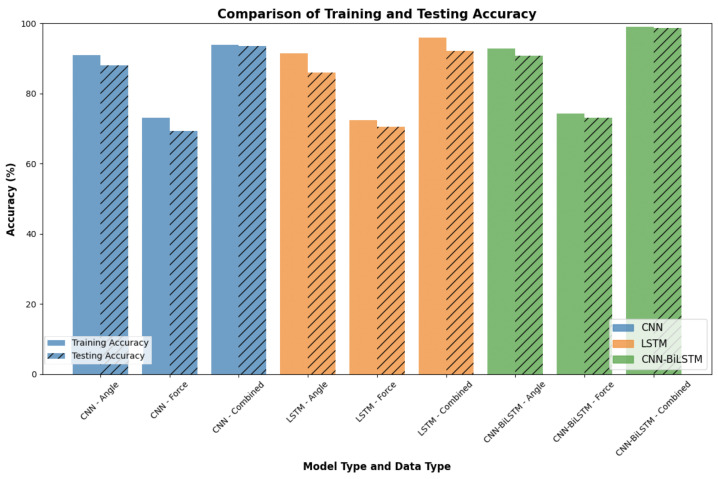
Training and testing accuracy comparision across different models.

**Table 1 sensors-24-04378-t001:** Top 16 configurations during ablation study.

Config.	Conv1D Filters	Conv1D Kernel Size	LSTM Units	Dense Units	Dense Units Softmax	Dropout Rate	Accuracy (%)
C1	32	2	32	32	8	0.1	98.2292
C2	32	2	32	32	8	0.5	98.3333
C3	32	2	32	64	8	0.1	98.4500
C4	32	2	32	64	8	0.5	96.8750
C9	32	3	32	32	8	0.1	95.9375
C11	32	3	32	64	8	0.1	96.5625
C13	32	3	64	64	8	0.1	95.8333
C14	32	3	64	64	8	0.5	94.5833
C16	32	3	64	32	8	0.5	95.2083
C18	64	2	32	32	8	0.5	95.8333
**C22**	**64**	**2**	**64**	**32**	**8**	**0.5**	**98.7542**
C24	64	2	64	64	8	0.1	98.6458
C26	64	3	32	32	8	0.5	94.6875
C28	64	3	64	32	8	0.1	97.1875
C29	64	3	64	64	8	0.5	96.8750
C31	64	3	64	64	8	0.5	96.3542

Note: Bold text denotes the optimal CNN-BiLSTM configuration, which is C22.

**Table 2 sensors-24-04378-t002:** Classification report for angle data.

Grasp Type	Precision	Recall	F1-Score	Accuracy (%)
Pulp Pinch (PP)	0.84	0.63	0.72	63.33
Lateral Pinch (LP)	0.88	1.00	0.94	100.00
Diagonal Volar Grip (DVG)	1.00	0.91	0.95	90.83
Cylindrical Grip (CG)	0.73	0.88	0.80	87.50
Extension Grip (EG)	0.95	1.00	0.98	100.00
Tripod Pinch (TP)	0.94	0.96	0.95	95.83
Spherical Grip (SG)	1.00	1.00	1.00	100.00
Hook Grip (H)	0.96	0.89	0.92	89.17
Accuracy (Average)				90.83%

**Table 3 sensors-24-04378-t003:** Classification report for force data.

Grasp Type	Precision	Recall	F1-Score	Accuracy (%)
Pulp Pinch (PP)	0.74	0.56	0.64	55.83
Lateral Pinch (LP)	0.96	0.67	0.79	66.67
Diagonal Volar Grip (DVG)	0.77	0.80	0.78	80.00
Cylindrical Grip (CG)	0.61	0.77	0.68	76.67
Extension Grip (EG)	0.71	0.63	0.67	63.33
Tripod Pinch (TP)	0.63	0.99	0.77	99.17
Spherical Grip (SG)	0.74	0.65	0.69	65.00
Hook Grip (H)	0.86	0.78	0.82	78.33
Accuracy (Average)				73.12%

**Table 4 sensors-24-04378-t004:** Classification report for combined data.

Grasp Type	Precision	Recall	F1-Score	Accuracy (%)
Pulp Pinch (PP)	0.97	0.99	0.98	99.17
Lateral Pinch (LP)	0.95	1.00	0.98	100.00
Diagonal Volar Grip (DVG)	1.00	0.97	0.98	96.67
Cylindrical Grip (CG)	1.00	0.97	0.98	96.67
Extension Grip (EG)	0.99	0.99	0.99	99.17
Tripod Pinch (TP)	1.00	0.99	1.00	99.17
Spherical Grip (SG)	1.00	1.00	1.00	100.00
Hook Grip (H)	0.99	0.99	0.99	99.17
Accuracy (Average)				98.75%

**Table 5 sensors-24-04378-t005:** Mean accuracy and standard deviation for different data types and train–test split ratios.

Data Type	Train–Test Split Ratio	Accuracy ± Standard Deviation (%)
Angle Data	60:30	84.09 ± 1.68
Angle Data	70:20	87.19 ± 1.75
Angle Data	80:10	90.83 ± 0.37
Force Data	60:30	69.15 ± 1.68
Force Data	70:10	71.36 ± 1.58
Force Data	80:10	73.12 ± 1.88
Combined Data	60:30	92.39 ± 0.77
Combined Data	70:20	96.73 ± 0.73
Combined Data	80:10	98.75 ± 0.80

**Table 6 sensors-24-04378-t006:** Comparison of different classification networks for the dataset.

Model	Time (s)	Accuracy (%)
SVM	0.42	89.38
Logistic Regression	0.61	93.67
Random Forest	1.39	94.36
KNN	0.22	88.38
ANN	0.31	91.33
CNN	6.20	93.51
LSTM	7.87	92.19
BiLSTM	4.92	93.85
CNN-LSTM	5.21	96.12
**CNN-BiLSTM (** * **Glove-Net** * **)**	**6.56**	**98.75**

**Table 7 sensors-24-04378-t007:** Comparison of classification algorithms from literature.

Authors	Classification Algorithm	Number of Movements	Accuracy (%)
Nassour et al. [33]	Linear Regression	15	89.4
Chen et al. [27]	SVM	16	89.4
Ayodele et al. [4]	CNN	6	88.3
Maitre et al. [34]	Random Forest	8	95
Maitre et al. [45]	CNN-LSTM	13	93
Lin et al. [35]	Linear Regression	3	98.0
Zhang et al. [3]	RBFNN	8	93.3
Calado et al. [29]	ANN	10	93.9
Dutta et al. [30]	SVM	19	92.0
Zheng et al. [23]	DBDF	52	93.15
**Proposed Work**	**CNN-BiLSTM**	**24**	**98.75**

## Data Availability

Data will be made available by the authors upon request.

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
