# Peer review of "Glove-Net: Enhancing Grasp Classification with Multisensory Data and Deep Learning Approach"

_sensors, 2024, doi:10.3390/s24134378_

Round 1
Reviewer 1 Report
Comments and Suggestions for Authors
In this manuscript, the authors proposed a novel methodology utilizing a multi-sensory data glove to capture intricate grasp dynamics, including finger posture bending angles and fingertip forces. Through rigorous numerical experimentation, the results underscore the significance of multimodal grasp classification and highlight the efficacy of the proposed hybrid Glove-Net architectures in leveraging multi-sensory data for precise grasp recognition.
I consider that the proposal is useful for grasp classification for understanding human interactions with objects. Some minor comments are:
1. In lines 38 to 45 of the article, the author presents the advantages and disadvantages of gesture capture and classification based on the field of vision, but the literature provided is all research in the field of wearable devices, which is somewhat subjective and negative, and it is recommended to read review articles based on applications in the field of vision to make a judgment.
2. In section 2, there is a lack of literature on the application of hand grasp categorization and force sensory feedback in it related to the topic of the article, and it is recommended to increase the research literature on these two parts, otherwise this will be somewhat questionable on the theoretical and practical basis of the topic of this article.
3. In Section 3.2, the authors do not state how many datasets were collected and whether they meet the minimum requirements, which are essential during the training of the neural network, and which will affect the assessment of the accuracy and truthfulness of the output results of the network model.
4. In section 3.2, the author's application of the two hybrid models, CNN and LSTM, should elaborate on how to combine or fuse them to form a new network model and explain the principles and processes, rather than presenting the two models individually, which can be troubling to the understanding of the new model.
5. This paper does not seem to specifically represent comparative data with other deep learning networks trained on the same dataset, and different experimental comparisons would enhance the persuasiveness of this paper as well as the rationality of the innovation of the new model.

Author Response
We have addressed all the concerns raised by the reviewer. Please see the attachment having responses to the reviewer's comments.

Reviewer 2 Report
Comments and Suggestions for Authors
see the file.

Author Response

(The authors gave the same response as above.)
